# Friends in Arms: Flavonoids and the Auxin/Cytokinin Balance in Terrestrialization

**DOI:** 10.3390/plants12030517

**Published:** 2023-01-23

**Authors:** Jasmina Kurepa, Timothy E. Shull, Jan A. Smalle

**Affiliations:** Plant Physiology, Biochemistry, Molecular Biology Program, Department of Plant and Soil Sciences, University of Kentucky, Lexington, KY 40546, USA

**Keywords:** flavonoids, auxin, cytokinin, oxidative stress, antioxidants, terrestrialization, shoot/root ratio, symbiosis

## Abstract

Land plants survive the challenges of new environments by evolving mechanisms that protect them from excess irradiation, nutrient deficiency, and temperature and water availability fluctuations. One such evolved mechanism is the regulation of the shoot/root growth ratio in response to water and nutrient availability by balancing the actions of the hormones auxin and cytokinin. Plant terrestrialization co-occurred with a dramatic expansion in secondary metabolism, particularly with the evolution and establishment of the flavonoid biosynthetic pathway. Flavonoid biosynthesis is responsive to a wide range of stresses, and the numerous synthesized flavonoid species offer two main evolutionary advantages to land plants. First, flavonoids are antioxidants and thus defend plants against those adverse conditions that lead to the overproduction of reactive oxygen species. Second, flavonoids aid in protecting plants against water and nutrient deficiency by modulating root development and establishing symbiotic relations with beneficial soil fungi and bacteria. Here, we review different aspects of the relationships between the auxin/cytokinin module and flavonoids. The current body of knowledge suggests that whereas both auxin and cytokinin regulate flavonoid biosynthesis, flavonoids act to fine-tune only auxin, which in turn regulates cytokinin action. This conclusion agrees with the established master regulatory function of auxin in controlling the shoot/root growth ratio.

## 1. Introduction

The plant hormones auxin and cytokinin act synergistically or antagonistically to regulate development and alter growth in response to environmental cues [1,2,3,4,5]. Auxin and cytokinin biosynthesis, transport, and signaling and their functional interactions have been extensively reviewed [3,6,7,8,9,10,11,12,13,14,15,16,17].

Auxin/cytokinin interactions control the shoot/root growth ratio, which is an essential adaptation for establishing the equilibrium in a terrestrial habitat characterized by water and mineral nutrient fluctuations [3]. For example, auxin action increases in response to decreased water availability and suboptimal mineral nutrient content of soils [3]. This increased auxin action leads to the development of a larger root system, which increases water and mineral nutrient uptake, and a smaller shoot, which reduces the consumption of water and mineral nutrients [3]. In contrast, when water and nutrient levels are optimal, cytokinin action increases, driving shoot growth while suppressing root growth and thus maximizing the carbon allocation for reproductive development. This auxin/cytokinin interaction appears to be unidirectional, with auxin inhibiting cytokinin biosynthesis and action [3]. Auxin action is upregulated under conditions where vigorous shoot growth is potentially lethal to the plant. Once auxin-enhanced root development and shoot growth inhibition reestablish homeostasis, auxin action declines, releasing cytokinin to promote shoot growth.

Although reproduction is key to the survival of a species, it needs to be under firm negative control to ensure the survival of the individual plant until it reaches the reproductive stage. The emerging picture is that the overarching effects of auxin action are aimed at an individual plant’s survival and well-being, whereas the overarching effect of cytokinin action is reproduction, as cytokinin ensures the maximal development of shoot organs and seed production (Figure 1). There is compelling evidence that this antagonistic auxin/cytokinin control of shoot/root growth ratios is operational in all land plants, including monocots [18,19,20,21,22] and primitive plants [3], for which the connection with nutritional status awaits further investigation.

The evolution of land plants coincided with a dramatic expansion in the complexity of secondary metabolic pathways, many of which play important roles in adaptation to environmental stresses [23,24]. The classical definition of plant metabolites follows a functional classification. It states that primary metabolites are found in all plants and are directly involved in normal growth, development, and reproduction. In contrast, secondary metabolites are often limited to a specific taxonomic group, and their function in normal growth and development is not apparent [25]. A more recent system of classification incorporates hormones as a third category of metabolites and discusses the increasingly vague boundaries between these functional groups as knowledge accumulates [23]. Phenolics are currently the largest class of secondary metabolites in plants, likely due to their dramatic expansion during terrestrialization [24]. Within this class, flavonoids are noteworthy for their influence over how the auxin/cytokinin hormone pair controls plant growth and physiology. Here, we provide an overview of the many interactions between flavonoids and auxin/cytokinin, ranging from the control of root and shoot development and physiology to engagement with beneficial symbionts. The summarized evidence supports the co-evolutionary relationship between plant hormones and flavonoids, highlighting that stress-responsive flavonoids primarily affect the action of auxin, the hormone of well-being.

## 2. Flavonoid Biosynthesis and Functions

### 2.1. The Flavonoid Biosynthetic Pathway

Flavonoids are a subclass of secondary metabolites synthesized by the phenylpropanoid biosynthetic pathway. Phenylpropanoids are a functionally diverse class of plant natural products with a vast assortment of species-specific permutations [26]. Regardless of its downstream variability, the core phenylpropanoid pathway begins with the deamination of phenylalanine to form *trans*-cinnamic acid by phenylalanine ammonia-lyase. Next, the activity of cinnamate 4-hydroxylase converts *trans*-cinnamic acid into *p*-coumaric acid. 4-coumaroyl CoA-ligase then converts *p*-coumaric acid into *p*-coumaroyl CoA, which can be transformed into various flavonoids or lignin monomers. Due to their roles as plant pigments and epidemiological evidence of their beneficial effect on human health, the biosynthesis of flavonoids is one of the most frequently studied branches of phenylpropanoid biosynthesis [27,28,29]. The first committed step to flavonoid biosynthesis is the conversion of *p*-coumaroyl CoA into naringenin chalcone by chalcone synthase, which is subsequently transformed into naringenin by chalcone isomerase. Naringenin is the precursor to antioxidant and defensive compounds of the flavonol and anthocyanin subclasses of flavonoids [30]. *p*-coumaryl-CoA is also converted into *p*-coumaraldehyde by the enzyme cinnamoyl-CoA reductase or into caffeic acid by *p*-coumarate 3-hydroxylase. Both compounds are precursors for the phenylpropanoid monomers that make up lignin [31]. Several gold-standard reviews comprehensively discuss phenylpropanoid and flavonoid biosynthetic pathways, their regulation, and species-specific distributions of different phenylpropanoid species [27,30,32,33,34,35,36,37,38,39,40,41].

### 2.2. Evolutionary Origin of Flavonoid Biosynthesis

Flavonoids co-emerged with land plants and are considered an essential adaptation to survive the terrestrial habitat [24,42]. It has been proposed that flavonoids originally served as a functional replacement for mycosporine-like amino acids (MAAs), the main protectants against ultraviolet light in algae [24]. However, in contrast to MAAs that have potent UV light screening capacity, flavonoids prevent UV-induced free radical damage [37]. Flavonoid biosynthesis does not require nitrogen, an advantage for land plants, which are more exposed to nitrogen limitation than aquatic plants. The subsequent evolution of flavonoid pathways towards increased complexity followed the gene duplication and diversification of the P450 cytochrome family and the concomitant synthesis of a wide range of genus- and species-specific flavonoid species used by land plants to survive diverse abiotic and biotic stresses [43,44].

### 2.3. Functions of Flavonoids

The biological functions of flavonoids are generally divided into three main categories. First, flavonoids can directly modulate primary metabolic pathways [45]. For instance, *Arabidopsis thaliana* plants bearing lesions in early steps of the flavonoid pathway are depleted not only in flavonoids but also in the respiratory factor ubiquinone, whose deficiency can be restored by addition of the flavonoid kaempferol to growth media [46]. Labeling experiments revealed that kaempferol is incorporated into ubiquinone, indicating that flavonoids are integrated into primary plant metabolic processes. Second, flavonoids have essential functions in the interactions with other organisms [41,47,48,49]. Examples of these interactions range from the attraction of pollinators and frugivores and the establishment of symbiotic relations with fungi and bacteria, to the suppression of pathogen invasion [34,41,47,49,50]. Lastly, flavonoids play essential roles in the defense and adaptation to abiotic stresses, such as drought, heavy metals, salinity, high/low temperature, and UV radiation [33,34,51,52,53,54,55]. Considering the abiotic stress-protective function of flavonoids, we can distinguish between their antioxidative function and their functions as modulators of hormone action.

## 3. Antioxidative Function of Flavonoids

Exposure to environmental stress often leads to increased production of reactive oxygen species (ROS) such as superoxide, hydrogen peroxide, and nitric oxide, which—when present in excess—negatively impact plant growth and development by causing oxidative damage to proteins, lipids, and nucleic acids [56,57,58,59]. ROS are a normal by-product of plant metabolism, and under non-stress conditions, ROS levels are balanced by an array of antioxidant enzymes (e.g., superoxide dismutase (SOD), ascorbate peroxidase, catalase) and non-enzymatic antioxidants such as ascorbate and glutathione [60,61]. Under stress conditions, these classic antioxidant defenses are induced, and the cellular capacity to detoxify ROS and neutralize the threat to survival increases [56,62,63,64,65,66]. However, under high-stress conditions, genes encoding key flavonoid pathway enzymes are induced, and the resulting increase in flavonoid levels is believed to help plants overcome intense or prolonged stress, mainly by quenching ROS [32,33,67,68,69,70,71,72,73,74,75,76,77]. Flavonoid biosynthesis requires a substantial allocation of carbon flow [78] and needs to be tightly controlled to avoid the unnecessary consumption of photosynthesis products.

Initially, flavonoids were primarily viewed as UV light protectants. The current consensus is that the main role of flavonoids is to quench ROS generated by a wide range of stressors, including UV exposure. We can recognize two modes of flavonoid-dependent reduction in ROS levels in the cell: the direct and the indirect mode [79,80].

### 3.1. Direct Antioxidative Effect of Flavonoids

Flavonoids act as direct antioxidants by reducing radicals with a higher redox potential than themselves (e.g., superoxide, peroxyl, alkoxyl, and hydroxyl radical) [55,79,81,82,83,84]. This reaction neutralizes ROS and yields a less-reactive flavonoid radical that may react with a second flavonoid radical to form a stable quinone structure [79]. The direct antioxidant activity of different flavonoids has long been documented in vitro [79]. Although the importance of flavonoids as direct in vivo antioxidants is still debated [81,84], it has been amply demonstrated that flavonoids play an important role in defense against ROS excess [53]. For example, it was shown that feeding plants as diverse as Arabidopsis, tobacco, and duckweed with the flavonoid quercetin protects them from the oxidative stress induced by the superoxide-generating compound paraquat [85]. Moreover, transgenic lines that over-accumulate flavonoids have increased oxidative stress tolerance (e.g., [86,87,88]).

### 3.2. Indirect Antioxidative Effect of Flavonoids

An example of the indirect antioxidant action of some flavonoids is their capacity to chelate metals through their catechol moiety [79]. The chelation of redox-active metals (e.g., iron and copper) reduces ROS damage generated by metal-catalyzed Fenton and Haber–Weiss reactions, whereas the chelation of non-redox active metals such as cadmium reduces oxidative damage by preventing the depletion of glutathione (which is an elicitor of heavy metal-detoxifying phytochelatins) and the heavy metal-dependent inactivation of antioxidative enzymes [89,90,91]. Indeed, flavonoids accumulate in plants undergoing heavy metal stress, and the exogenous application of flavonoids partially protects plants from the effects of exposure to heavy metals [68,92].

## 4. Flavonoid-Dependent Modulation of the Auxin/Cytokinin-Regulated Shoot/Root Ratio

Current studies suggest a co-evolutionary relationship between auxin and cytokinin on one side and flavonoids on the other [3,24,42]. Here, we propose that during evolution, flavonoids also acquired a function of fine-tuning the auxin/cytokinin-dependent regulation of the shoot/root growth ratio to help ensure the well-being of the plant until it reaches reproductive maturity. Although flavonoids affect similar processes in shoots and roots (Figure 2), the distribution of flavonoid species and the function of auxin and cytokinin in roots and shoots is quite different, and we will present examples of the flavonoid-dependent modulation of hormone-regulated processes in these organs separately.

### 4.1. Roots

Analyses of flavonoid biosynthesis mutants provided the first evidence that flavonoids are involved in root development. Arabidopsis *tt4* mutants have a defect in the gene encoding chalcone synthase, the enzyme that catalyzes the first committed step in flavonoid synthesis [39,93]. These flavonoid-deficient mutants have a longer primary root than wild-type plants, increased lateral and adventitious root formation, and a higher root hair frequency, suggesting that flavonoids inhibit root growth [94]. Later studies showed that the accumulation of flavonols in root tips leads to decreased root elongation and that this effect is due to inhibited auxin transport [95]. The hormone gibberellin (GA) plays an important regulatory role in this process, as GA suppresses root tip flavonol accumulation by inhibiting the activities of specific MYB transcription factors that promote the expression of genes encoding flavonol pathway enzymes [95]. This inhibitory action of GA involves the accelerated degradation of DELLA proteins which activate the MYBs. As DELLA proteins are key mediators of how plants inhibit their growth in response to elevated stress conditions [96], they simultaneously promote oxidative stress tolerance [97], and this includes the promotion of flavonoid biosynthesis [95,98,99]. In roots, this DELLA-dependent mechanism likely helps fine-tune growth in response to environmental stresses, with flavonoid induction providing oxidative stress tolerance and controlling auxin transport. Moreover, DELLA proteins modulate the oxidative stress status and development of lateral roots and root hairs, implying that this environmental input mechanism impacts the entire root system [97,100].

Studies using different flavonoid pathway mutants defective in enzymatic steps downstream of chalcone synthase, combined with flavonoid feeding experiments, revealed that flavonols, particularly kaempferol, are also responsible for suppressing lateral root formation [101]. The current view is that kaempferol quenches auxin-induced ROS generated to alter the cell wall consistency and allow for the emergence of the lateral roots [101]. Auxin also promotes flavonol synthesis in lateral root formation zones [102]. By upregulating flavonol biosynthesis, auxin both limits its action in shaping the root system and protects the root from excessive ROS accumulation. This auxin-ROS-flavonol loop also has an interesting implication for auxins’ role in responding to nutrient deficiency. Auxin action increases in response to low levels of nitrogen and phosphorus in soils, which typically leads to increased lateral root initiation and growth to help plants secure more of these essential nutrients [3]. It was found that phosphorus deficiency leads to decreased root content of the flavonols kaempferol and quercetin and their derivatives, which in turn promotes lateral root emergence [103]. Because phosphate deficiency is associated with increased auxin transport from the shoot apex and with increased auxin activity in roots, flavonols might also be involved in the regulation of auxin transport and action in phosphate-deficient plants [103].

Similar to the role of the flavonol kaempferol in limiting auxin-induced lateral root emergence, root hair formation (also an auxin-induced process) is negatively impacted by the flavonol quercetin [104]. Quercetin has a dual function in this regulatory step: first, it quenches the auxin-induced ROS that promotes root hair initiation and growth, and second, it controls auxin action through the inhibition of auxin transport and auxin activity. In the absence of quercetin, auxin transport from the root tip is increased, leading to increased root hair formation, which is further enhanced by auxin-induced ROS accumulation that is not counteracted by the quercetin-dependent sequestration of ROS.

Flavonoid-deficient mutants are also defective in root gravitropic and light-avoidance responses [105,106]. Tropic responses involve the unequal distribution of auxin, which leads to the differential expansion of cells across an organ and results in bending growth [107]. Flavonols also accumulate unequally across the root bending region, leading to the unequal inhibition of auxin transport and a root gravitropic response [108]. An unequal distribution of flavonols in the root elongation zone leads to the asymmetrical distribution of PIN-FORMED (PIN) auxin efflux carriers, resulting in the asymmetrical auxin distribution required for differential growth. Flavonols also play an essential role in the light-avoidance response, a process that involves both auxin and cytokinin [106]. Roots typically develop in the soil’s dark environment and respond to light by bending away from the light source. Flavonols (quercetin and kaempferol) accumulate at the side of roots exposed to the light source and lead to the cell elongation of the illuminated region. In parallel, light- and cytokinin signaling-dependent flavonol biosynthesis leads to the accumulation of flavonols in the root meristem and reduces the meristem size and cell proliferative root growth. The latter effect is likely caused by the suppression of auxin action through the increased quenching of auxin-induced ROS. The combination of these two flavonol-mediated effects results in bending growth away from the light source [106].

Since the biosynthesis of flavonoids is not only hormone-regulated but also controlled by a plethora of environmental cues [30,37,39], flavonols seem to act as transducers of environmental changes to the auxin pathway that regulates the development of roots and, thus, allows for a more flexible adaptation to soil conditions.

### 4.2. Shoots

Flavonoids of the anthocyanin, flavonol, and isoflavonoid classes have a significant impact on auxin- and cytokinin-mediated shoot growth and physiology. Resembling their functions in roots, flavonoids interact with auxin in shoots to control the leaf petiole angle and leaf nastic growth [109,110]. The control of the leaf petiole angle is a process that adjusts the leaf blade orientation to the light source and thus maximizes photosynthesis [111]. In soybean, the leaf petiole angle is controlled by the asymmetric distribution of isoflavonoids which mirrors the direction and the intensity of the light source [110]. This asymmetric distribution leads to the asymmetric expression of *PIN1*, leading to an accumulation of auxin in cells opposite the light source. Interestingly, the loss of PIN function in soybean and Arabidopsis leads to opposite effects on the leaf petiole angle [110]. The adaptive function of this variability of auxin- or flavonol-dependent leaf petiole angle control in different plant species is not yet understood. Related to the leaf petiole angle control, the hyponastic (upward) and epinastic (downward) bending of leaves are essential for shoot architectural plasticity [111], and flavonols again play an important role in its control. For example, Arabidopsis *rol1* mutants have altered flavonol glycosylation which leads to an increased level of kaempferol derivatives [109]. The accumulation of kaempferol derivatives and a concomitant increase in the inhibition of auxin transport in these mutants causes their characteristic hyponastic leaves phenotype [109].

In addition to these specific developmental controls, there is evidence for more extensive regulation of shoot development by the auxin/flavonoid interaction. Both auxin and flavonoid biosynthesis in the shoot are light-induced. In addition, auxin transport and flavonoid accumulation patterns largely overlap. This spatial and stimulus-related co-distribution suggests functional co-dependence. Indeed, transgenic plants that hyper-accumulate flavonoids are severely dwarfed due to the increased inhibition of auxin transport throughout the plant [109].

Anthocyanins are a major class of shoot flavonoids, and their function relevant for this review is the quenching of stress-generated ROS [86]. For example, drought and nutrient deficiency leads to a substantial increase in anthocyanin content in the shoot [112,113,114,115,116]. It is now accepted that this response protects the shoot from increased ROS accumulation associated with both types of stresses [112,113,114,115,116]. Cytokinin treatment suppresses this response, and strong cytokinin-resistant mutants have higher anthocyanin content and increased drought tolerance [117]. Drought stress is known to downregulate the cytokinin action to attenuate shoot growth and decrease the stomatal conductance to help plants survive water deficiency [3].

However, early studies showed that cytokinin induces anthocyanin biosynthesis, a finding confirmed and strengthened by transcriptomics analyses and the identification of the complement of cytokinin-induced genes that encode components of the anthocyanin biosynthesis pathway [118,119]. More recent research showed that the cytokinin-induction of anthocyanin accumulation is strictly light-dependent and enhanced by sucrose [120]. This light-dependency involves the blue part of the spectrum and depends on and is correlated with the rate of photosynthetic electron transport, indicating that this anthocyanin synthesis induction serves to ameliorate photosynthetic ROS generation [121]. Indeed, higher photosynthetic rates caused by exposure to elevated CO_2_ give rise to higher anthocyanin biosynthesis, which—under high-CO_2_ conditions-–can be sustained by the increased rate of carbon fixation [121].

These apparently contradictory effects of cytokinin on anthocyanin synthesis in shoots can be resolved by considering the overall levels of anthocyanin accumulation, which are high in response to drought and nutrient deficiency and moderate under optimal water and nutrient availability conditions. In conditions that allow vigorous shoot growth, accompanied by high photosynthesis rates and high generation of ROS, cytokinin both promotes anthocyanin synthesis to safeguard cellular integrity and limits it to a level that is not excessively costly in terms of carbon flow allocation [122]. From this perspective, it follows that cytokinin action is downregulated in response to drought and nutrient deficiency, which leads to shoot growth inhibition and increased anthocyanin accumulation, which serves as an additional protection against the damaging effects of stress-generated ROS.

In shoots, interactions between flavonoids and the auxin/cytokinin hormone pair are consistent with the proposed individual survival and well-being function of auxin and the reproduction-focused function of cytokinin. While cytokinin promotes anthocyanin synthesis to a level needed for ROS protection, it simultaneously limits the carbon flow through this pathway to ensure maximal carbon availability for shoots and, thus, reproductive development. In addition, the flavonoid/auxin interactions in the shoot are subjected to light control, thus connecting the auxin action to light conditions, and thus ensuring the optimal developmental adaptation of shoots to the light environment.

## 5. Flavonoid-Auxin/Cytokinin Link and Symbiosis

Possibly the best example of flavonoids supporting and modulating the auxin/cytokinin-regulated shoot/root ratio (and, through that, supporting the plant reproduction/well-being balance) are the symbiotic interactions between plants and beneficial bacteria and fungi. The overarching theme is that flavonoids act together with auxin to promote symbiotic interactions, whereas cytokinin—albeit required for some symbiotic development—limits symbiosis to the level adequate for survival, thus ensuring sufficient but not excessive allocation of carbon for this process.

### 5.1. Symbiosis with Arbuscular Mycorrhizal Fungi (AMF)

AMF symbiosis was one of the major steps in the evolution of land plants [123]. By interacting with plant roots, the fungal hyphae serve as an extension of the root system, helping plants with water and mineral nutrient uptake [124]. In turn, the fungus receives carbohydrates from the plant, creating a mutually beneficial relationship. AMF symbiosis is widespread and is estimated to involve up to 80% of plant species [125,126].

Both auxin and flavonoids play a positive role in AMF symbiosis [127]. Root-secreted flavonoids have been shown to promote fungal spore germination, hyphal growth, and the fungal colonization of roots [128]. Subsequent research showed a positive correlation between fungal root colonization rates and flavonoid biosynthesis and secretion levels [129]. In addition, fungal inoculation itself leads to the increased synthesis of flavonoids, a process that was shown to be important for successful root colonization and likely based on the flavonoid-dependent inhibition of auxin transport [24]. The fungal interaction also increases auxin synthesis and action, and since this symbiotic interaction is defective in auxin-deficient and auxin-insensitive mutants, this increased auxin action is required for root colonization [130]. Contrary to the positive effects of flavonoid and auxin, cytokinin plays a largely negative role in AMF symbiosis. The establishment of AMF symbiosis leads to increased cytokinin biosynthesis [131], and this is thought to decrease root growth in favor of increased shoot growth, which is a distinctive response to increased nutrient availability and further evidence that cytokinin acts as a satiation hormone [3]. In support of that hypothesis, cytokinin was shown to suppress root penetration and colonization by the fungus [132] as well as hinder carbon-feeding of the fungus by the plant, thus restricting the colonization of roots to levels that combine optimal nutrient uptake with the optimal distribution of carbon between the shoots and roots [133].

AMF symbiosis plays a central role in the acquisition of phosphate from soils [134]. Importantly, soil phosphate content is a major deciding factor in the rate of AMF symbiosis, as high phosphate content tends to suppress this plant–fungal interaction [135]. Phosphate deficiency, on the other hand, leads to a combined increase in auxin action and decreased cytokinin biosynthesis and sensitivity [3], which generates an optimal hormonal balance for the fungal colonization of the roots (Figure 3).

### 5.2. Rhizobial Symbiosis

Nitrogen availability is a major limiting factor for plant growth, and the ability of some plant species to engage in symbiosis with the nitrogen-fixing bacteria of the Rhizobium genus has provided them with the substantial benefit of having an additional source of ammonium and, thus, nitrogen [136,137,138]. In this symbiotic interaction, the bacteria are housed in nodules, specialized root structures whose initiation and development are triggered by signal exchanges between the plant and bacterium. As with AMF, the Rhizobium symbiont benefits by acquiring photosynthates from the plant.

Flavonoids play multiple roles in Rhizobial symbiosis [139]. While flavonoids’ role as bacterial chemoattractants has been recently questioned [140,141,142], it is well established that, under nitrogen-limiting conditions, legumes secrete flavonoids as root exudates to promote the activation of NodD transcription factors. The activation of NodDs induces the expression of Rhizobia Nod factors, subsequently signaling for the plant to engage in nodule development [139,140,143]. In turn, Rhizobial infection induces flavonoid synthesis, which is important for modulating auxin action in infected roots, further induction of bacterial Nod factor gene expression, and ensuring host-specific bacterial interactions [47,144]. Similarly to AMF symbiosis, auxin plays an overall positive role in Rhizobial symbiosis by promoting nodule formation, a process that involves auxin transport inhibition by flavonols [145]. Cytokinin signaling is also required to establish Rhizobial symbiosis and functions, mainly by promoting the flavonoid biosynthesis needed for auxin transport control [145,146,147]. However, the overall effect of cytokinin appears to be the suppression of nodulation frequency, similar to its satiation hormone activity in AFM symbiosis (Figure 4). Rhizobium symbiosis leads to increased biosynthesis of cytokinin in shoots, followed by cytokinin transport to the roots, which reduces nodulation events [148]. A recent study shows that this suppression of nodulation requires type-B response regulators, which are essential components of the canonical cytokinin response pathway [149]. Thus, like AMF symbiosis, cytokinin suppresses further symbiotic engagement (i.e., nodulation) when the nutrient uptake is sufficient to favor an increase in the shoot/root growth ratio.

## 6. Conclusions

In addition to playing essential roles in providing stress tolerance to plants, evidence has accumulated in support of a co-evolutionary role for flavonoids and auxins in promoting the well-being (i.e., environmental adaptability) of land plants. Through their environmentally regulated biosynthesis and control of auxin transport and action, flavonoids ensure optimal auxin regulation for plant environmental adaptation. These auxin/flavonoid interactions agree with the master regulatory role of auxin in shaping plant development, and they coordinate protection from stress with shoot and root development, thus allowing land plants to occupy the challenging and highly variable conditions of the terrestrial environment. Although cytokinin also regulates flavonoid synthesis, its action is not directly regulated by flavonoids. Instead, flavonoids, especially flavonols, indirectly control cytokinin by regulating auxin action.

Although this review has focused on the interactions between flavonoids and the auxin/cytokinin hormone pair, flavonoids also interact with other plant hormones to further increase land plant adaptability [36,95,150,151,152,153,154]. While hormone-regulated flavonoid synthesis is at the base of most of these interactions, the interaction of flavonoids and abscisic acid (ABA) stands out as flavonols regulate ABA signaling, which aligns with the essential role that ABA plays in plant survival under adverse conditions [36]. Together with the key roles played by flavonoids in pathogen resistance [155,156], in the repelling of herbivores [157], and in the attraction of pollinators [158,159], the emerging picture is that the flavonoids, rather than being mere secondary metabolites, are truly key players in how plants have mastered the terrestrial environment.

## Figures and Tables

**Figure 1 plants-12-00517-f001:**
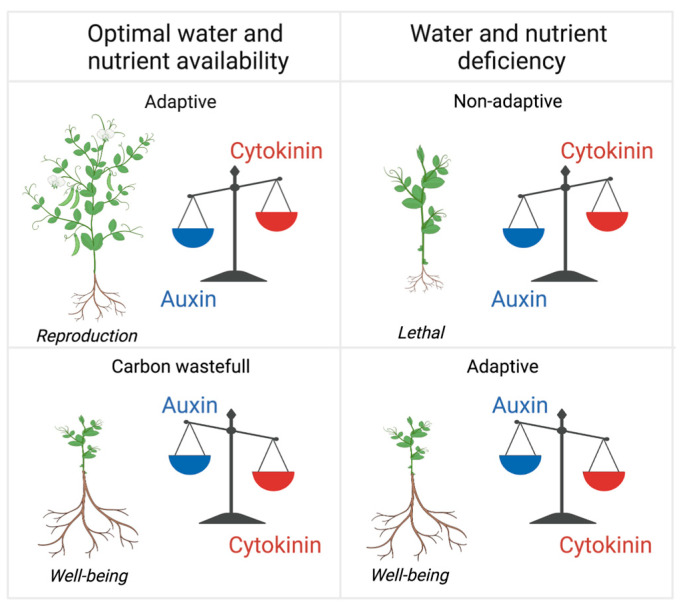
The auxin/cytokinin ratio and water and nutrient availability. Under optimal water and nutrient availability, auxin action decreases with a concomitant increase in cytokinin action, which increases the shoot/root growth ratio and leads to reproduction. A high auxin/cytokinin ratio would be suboptimal under these conditions, as it would lead to the needless allocation of carbon to root growth. Under water and nutrient deficiency conditions, auxin action is increased and cytokinin action is decreased, which ensures plant survival and reproduction, albeit at a lower rate. A high cytokinin/auxin ratio would be lethal under these conditions; vigorous shoot growth would cause potentially deadly water loss, and the insufficient allocation of carbon flow to roots would impede an increase in root development needed to acquire adequate amounts of water and nutrients.

**Figure 2 plants-12-00517-f002:**
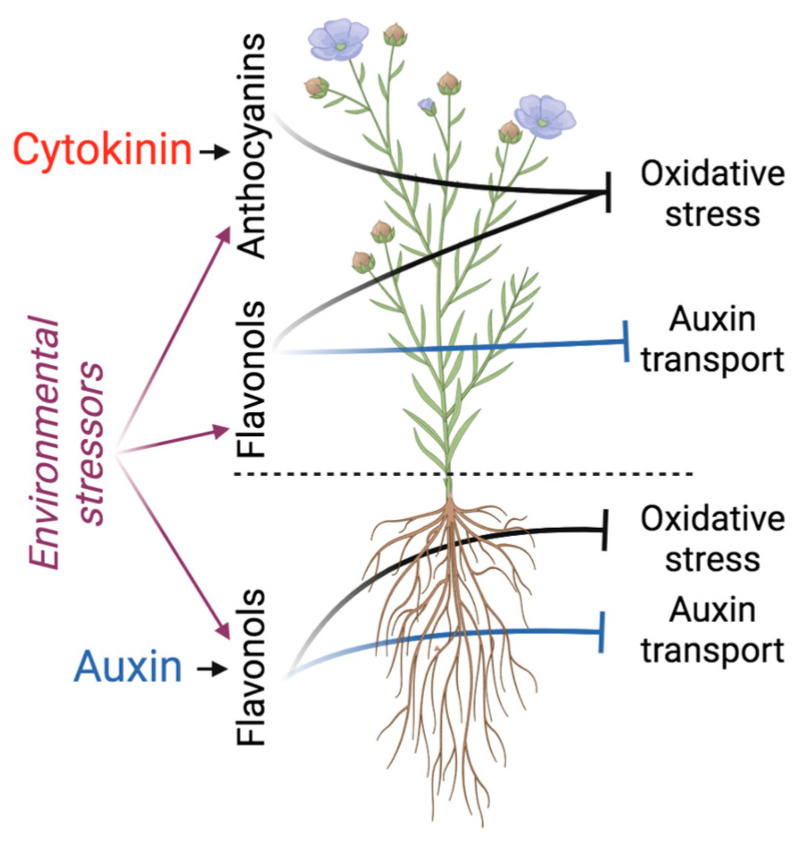
Auxin and cytokinin interactions with flavonoids in shoots and roots. Cytokinin, the promoter of shoot growth, upregulates the biosynthesis of anthocyanins in shoots, strengthening the antioxidative system poised to combat the excess of ROS generated, for example, by photosynthesis or by the action of environmental stressors. Auxin in roots upregulates the biosynthesis of flavonols, which modify auxin action by inhibiting auxin transport and counteracting auxin-induced ROS accumulation. The flavonoid-dependent inhibition of auxin transport is not restricted to roots—it also acts in shoots to shape petiole angle and leaf nastic development. In addition, flavonol synthesis is environmentally controlled, thus linking environmental fluctuations to changes in auxin action for optimal environmental adaptation.

**Figure 3 plants-12-00517-f003:**
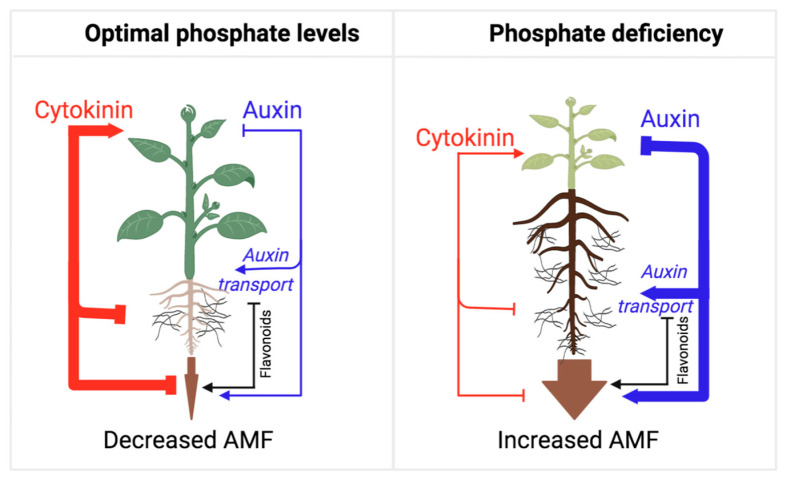
Model illustrating the auxin/cytokinin ratios during the establishment of AMF symbiosis from the perspective of adaptive control of the shoot/root growth ratio. As long as the plant perceives phosphate deficiency in the soil, the rate of AMF symbiosis establishment is high, mirroring a low shoot/root growth ratio and the allocation of resources towards root growth and fungal colonization. Under phosphate-deficient conditions, auxin action is high, which represses cytokinin action and shoot growth in favor of root growth and symbiosis. Under phosphate-sufficient conditions, auxin action is reduced, and cytokinin action is released to promote shoot growth, inhibit root growth, and suppress AMF symbiosis. Flavonoids play an essential role in this process, as they promote several stages of this symbiotic interaction.

**Figure 4 plants-12-00517-f004:**
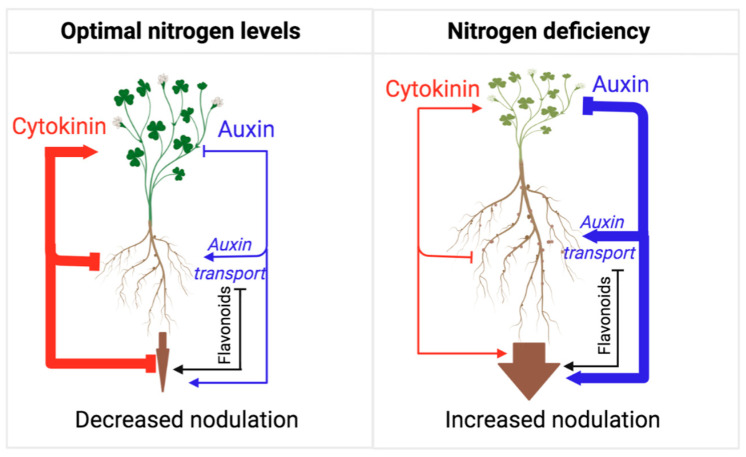
Model representing the actions of auxin and cytokinin on root nodulation from the perspective of adaptive control of the shoot/root growth ratio. As long as the plant perceives nitrogen deficiency in the soil, nodulation frequency is high, mirroring a low shoot/root growth ratio and the allocation of resources towards root growth and nodulation. Under nitrogen-deficient conditions, auxin action is high, which represses cytokinin action and shoot growth in favor of root growth and nodulation. Under nitrogen-sufficient conditions, auxin action is reduced, and cytokinin action is released to promote shoot growth and inhibit root growth and nodulation. This mechanism ensures optimal resource allocation for shoot growth and reproduction under optimal nitrogen conditions. Flavonoids play an essential role in this process, as they promote several stages in this symbiotic interaction and thus help with the auxin-dependent promotion of nitrogen fixation.

## Data Availability

Not applicable.

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
