# Peer review of "Friends in Arms: Flavonoids and the Auxin/Cytokinin Balance in Terrestrialization"

_plants, 2023, doi:10.3390/plants12030517_

Round 1
Reviewer 1 Report
I would like to thank the authors for their efforts to do this interested review which describing the interaction between flavonoids and the auxin/cytokinin balance to control plant growth performance under nutrient availability or nutrient deficiency. There are interested issues in the current review which led me to judge the present manuscript as “acceptable”
Introduction is well written and explained the main aim of this work. It includes a figure that simply explain the complex interaction between flavonoids and the auxin/cytokinin balance in both root and shoot systems under nutrient availability or nutrient deficiency.
The second section of this review focused on the biosynthesis pathway of flavonoids and summarized the most important functions of flavonoids under biotic and abiotic stress.
As exposure to environmental stress often leads to the increased production of reactive oxygen species (ROS) such as superoxide (O2•-), hydrogen peroxide (H2O2), and nitric oxide (NO), which negatively impact plant growth. So, direct and indirect antioxidative effects of flavonoids were covered in the third section.
The main core of this review is section 4 which interestingly explained the complex interaction between flavonoids and the auxin/cytokinin regulation of Shoot/Root ratio. In addition, section 5 focused the role of that interaction in plan-mycorrhizal and plant-rhizobial symbioses.
Minor comment:
Numbering system of review section should be revised.
Section 5 which explains the role of flavonoids and the auxin/cytokinin regulation in plant-AMF or plant-rhizobial symbioses should be revised to include of most of recent aspects of that interaction.
Reviewer 2 Report
Dear author(s),
there are some inspiring insights thorough this review manuscript and I tend to agree on its publication. However, there are also few points that can be quickly addressed to improve its overall communication. Most importantly, let me kindly remind you that the quality of any review manuscript is not defined by the quantity of summarized papers but by the synthesis of reviewed literature and identyfing new challenges, industrially promising directions for future research etc.
1/ consider increasing the attractiveness of the Title by highlighting the new knowledge you have revealed based on the synthesis upon the reviewed literature
2/ better justify the urgency and significance to review existing literature (in the Abstract)
3/ with regards to the "Abstract", please understand that "Literature review" is not just about summarizing the up-to-date literature, you need to perform its synthesis and reveal something new, propose some improvements and direction for future research (what was firstly discovered by literature review? where is the originality?)
4/ remove all clusters of references to avoid reference overkill (prefer only 1 reference to support 1 claim)
5/ go straight to the point and more in depth, write more technically (always provide corresponding numbers), significantly condensate all the text by reducing ballast phrases and cliché
6/ deeper comment on nutrient availability, complexity of phoshorus availability to organisms should be better explained, refer to Fig. 1 in paper "Novel sorbent shows promising financial results on P recovery from sludge water"
7/ make sure that this chapter fully introduces any reader into to the topic, explain all the terms, units, abbreviations, Latin and Greek letters, and the whole context that is necessary for anyone (including experts from other disciplines) to understand the following chapters
8/ consider referring to papers dealing with water issues ("Spatial distribution of groundwater quality in connection with the surrounding land use and anthropogenic activity in rural areas", "The Destructive Impact of Burned Peatlands to Physical and Chemical Properties of Soil")
9/ the research hypothesis (and motivation for literature review) could be stated more clearly, condensate the research hypothesis into 1 short statement or question that will be subsequently confirmed or refuted, make sure the urgency and significance of the research hypothesis was justified in its environmental - economic nexus
10/ I´m missing more agricultural point of view, consider to better communicate the industrial importance of these findings (get inspired in papers "Silica nanoparticles from coir pith synthesized by acidic sol-gel method improve germination economics" and "Economic impacts of soil fertility degradation by traces of iron from drinking water treatment")
11/ reveal the main driving mechanisms of all the biosynthesis and functions, provide deeper synthesis and reveal some more original/significant findings
12/ originality of these "Conclusions" is questionable (some can be traced in reviewed literature), present only original and industrially significant revelations that have the potential to expand the horizon of human knowledge (higher level of synthesis over reviewed literature is mandatory)
13/ clearly indicate whether the research hypotheses tends to be confirmed or not
Reviewer 3 Report
The authors summarize the relevance of direct flavonoid/auxin interactions for terrestrialization of land plants and also point to a more indirect effect on cytokinins, conveyed by hormonal balance of auxin/cytokinin levels, essentially with an effect on shoot/root ratios. Overall the review is well written, supported by most recent references, just enough details to raise the curiosity of the reader. The authors do without complicated Figures, just a few schemes and models that represent the main conclusions of this review, sufficiently explained in the legends. No redundant, yet well know flavonoid pathway or complicated Figures that illustrate transcriptional or hormonal networks. Whoever wants to go into more detail can do this based on the more than 100 references.
The review to my opinion requires a few additional statements:
1. Please somewhere indicate that flavonoid biosynthesis is not only linked to auxins/cytokinins but to a complete hormonal network, e.g. flavonoids biosynthesis is induced by jasmonate specifically during flowering and ripening (increasing reproductive success), but also in root formation (e.g. Sun et al., Plant Cell 2009) in potential combination with ethylene This can be limited to one or two sentences (may be one or two references). Either later in the conclusive statements, or quite early in the introduction. It is obvious that this is not the major focus of this review but it should at least be mentioned.
The regulatory network is much more complicated than we think.
2. Chapter 2.2 should be extended by additional points. Flavonoid biosynthesis is not only a functional replacement of MAAs but also goes along with rapid diversification of mono- and oxygenases, specifically P450s resulting in the chemically partly equivalent decoration and storage of oxygen in the aromatic B-ring (in addition to +/- oxygen in the A and C-ring, flavone and flavonol), not necessarily linked to changes in the UV-spectra This not only eliminates oxygen from the atmosphere in a stable molecule but also offers the possibility for additional decorations, and subsequent specificity in organismal interactions.
3. Could you please comment if there are clear differences between angio- and gymnosperms in the auxin/cytokinin balance, also monocots/dicots?
4. Flavonoid synthesis "flows out" of the shikimate pathway…. Please rephrase
